# PathFusion: Path-consistent Lidar-Camera Deep Feature Fusion

## Abstract

Fusing camera with LiDAR is a promising technique to improve the accuracy of 3D detection due to the complementary physical properties. While most existing methods focus on fusing camera features directly with raw LiDAR point clouds or shallow 3D features, it is observed that direct deep 3D feature fusion achieves inferior accuracy due to feature mis-alignment. The mis-alignment that originates from the feature aggregation across large receptive fields becomes increasingly severe for deep network stages. In this paper, we propose PathFusion to enable path-consistent LiDAR-camera deep feature fusion. PathFusion introduces a path consistency loss between shallow and deep features, which encourages the 2D backbone and its fusion path to transform 2D features in a way that is semantically aligned with the transform of the 3D backbone. We apply PathFusion to the prior-art fusion baseline, Focals Conv, and observe more than 1.2% mAP improvements on the nuScenes test split consistently with and without testing-time augmentations. Moreover, PathFusion also improves KITTI $AP_{3D}$ (R11) by more than 0.6% on moderate level.

## 1 Introduction

LiDARs and cameras are widely used in autonomous driving to provide complementary information for 3D detection (e.g., Caesar et al., 2020; Sun et al., 2020; Geiger et al., 2013). LiDAR point clouds capture better geometry information but suffer from a low resolution due to the power and hardware limitations. By contrast, cameras capture dense and colored images with more semantic information but usually lack the shape and depth information for geometry reasoning. As a result, recent methods (Li et al., 2022c; Vora et al., 2020; Liang et al., 2022; Li et al., 2022b) propose to fuse 2D and 3D features from LiDAR point clouds and camera images to enable accurate and robust 3D detection.

LiDAR and camera fusion requires to project the features into the same space. Previous works have proposed to either lift 2D camera features into 3D space (e.g., Vora et al., 2020; Wang et al., 2021a; Li et al., 2022b; Chen et al., 2022c;a) or to align both 2D and 3D features in a common representation space like bird-eye-view (BEV) (e.g., Liu et al., 2022c; Liang et al., 2022).

One important question for feature fusion is to select the correct fusion stage and we illustrate popular choices in Figure 1. Shallow fusion in Figure 1 (a) (Chen et al., 2022a; Vora et al., 2020; Chen et al., 2022d) fuses camera features with raw LiDAR points directly or shallow LiDAR features. Although shallow fusion benefits from the LiDAR and camera calibration for better feature alignments, the camera features are forced to go through multiple modules which are specialized for 3D feature extractions instead 2D camera processing (Li et al., 2022b), e.g., voxelization. By contrast, deep fusion (Xu et al., 2021b; Li et al., 2022b) in Figure 1 (b) enables more dedicated LiDAR and camera feature processing. However, since multiple LiDAR points are usually voxelized together and further aggregated with neighboring voxels before fusion, one voxel may correspond to a number of camera features, making the feature alignment very ambiguous. The feature mis-alignment significantly degrades the overall network accuracy and forces the majority of existing works to choose shallow fusion (Chen et al., 2022a).

In this paper, we propose PathFusion to augment existing shallow fusion methods and enable fusing deep LiDAR and camera features. PathFusion introduces a novel path consistency loss to regularize the 2D feature extraction and projection and explicitly encourages the alignments between LiDAR

Figure 1: Overview of three different strategies to fuse the camera and LiDAR features: (a) shallow fusion accurately fuses the 2D feature to the shallow 3D feature; (b) deep fusion projects the 2D features to the 3D feature space; (c) our method proposes path consistency loss to mitigate the feature mis-alignment problem and enables augmentation of shallow fusion methods with deep fusion for better accuracy.

and camera deep features. As shown in Figure 1 (c), we construct different paths between 2D and 3D features and minimize the semantic differences of features from different paths to enforce feature alignments. PathFusion can be easily applied to prior-art fusion methods, e.g., Focals Conv (Chen et al., 2022a), and demonstrates consistent accuracy improvements. Compared to existing piror-of-the-art shallow fusion method, FocalConv-F, PathFusion achieves 0.62% $AP_{3D}$ (R11) improvement on KITTI and 1.2% mAP and 0.7% NDS improvement on nuScenes test set.

The rest of the paper is organized as follows. Section 2 introduces related work and section 3 describes the background of 2D and 3D feature fusion. Section 4 provides a motivation example on the challenge of deep feature fusion and we present our method in section 5. Section 6 summarizes our results.

## 2 RELATED WORK

**3D Object Detection with LiDAR or Camera** 3D object detection targets at predicting 3D bounding box and can be conducted based on 3D point clouds from LiDARs or 2D images from cameras. LiDAR-based methods mainly encode the raw LiDAR point clouds (Qi et al., 2017a;b), or the voxelized into sparse voxel (Zhou & Tuzel, 2018) to process the input source as multi-resolution features. Based on those 3D features, various single-stage, two-stage 3D detection heads (Shi et al., 2020; 2019; Deng et al., 2021; Bhattacharyya & Czarnecki, 2020; Yan et al., 2018; Yang et al., 2020; He et al., 2020; Zheng et al., 2021; Lang et al., 2019; Yin et al., 2021a; Yan et al., 2018; Shi et al., 2021; Wang et al., 2020) are proposed to predict the 3D bounding boxes of target objects.

Another series of works focus on camera-based method which encodes single-view or multi-view images with 2D backbones such as ResNet (He et al., 2016). Due to a lack of depth information, 2D-to-3D detection heads are devised to enhance 2D features with implicitly or explicitly predicted depth to generate the 3D bounding box (Liu et al., 2022a;b; Wang et al., 2021c; Huang et al., 2021; Reading et al., 2021; Xie et al., 2022; Huang & Huang, 2022; Li et al., 2022c; Wang et al., 2021b).

**LiDAR-camera Fusion** Because of the complementary properties of LiDARs and cameras, recent methods propose jointly optimizing both modalities and achieving superior accuracy compared to LiDAR- or camera-only methods. As in Figure 1 (a) and (b), these methods can be largely divided into two categories depending on the fusion stages: (a) shallow fusion decorates the point clouds or shallow LiDAR features with image features to enrich the LiDAR inputs with the image semantic prior (Vora et al., 2020; Yin et al., 2021b; Xu et al., 2021a; Chen et al., 2022c;d; Wu et al., 2022; Li et al., 2022a; Chen et al., 2022a;b; Liang et al., 2018a); (b) deep fusion lifts image features into 3D space and combines them in the middle or deep stages of the backbone (Liang et al., 2018b; Huang et al., 2020).

Shallow fusion methods, e.g., Focals Conv (Chen et al., 2022a), LargeKernel3D (Chen et al., 2022b), have achieved state-of-the-art accuracy while deep fusion models suffer from the increasingly severe mis-alignment between camera and LiDAR features. Recently, Transfusion (Bai et al., 2022) and DeepFusion (Li et al., 2022b) propose to align the LiDAR and camera features with transformer and leverages cross-attention to dynamically capture the correlations between image and LiDAR features

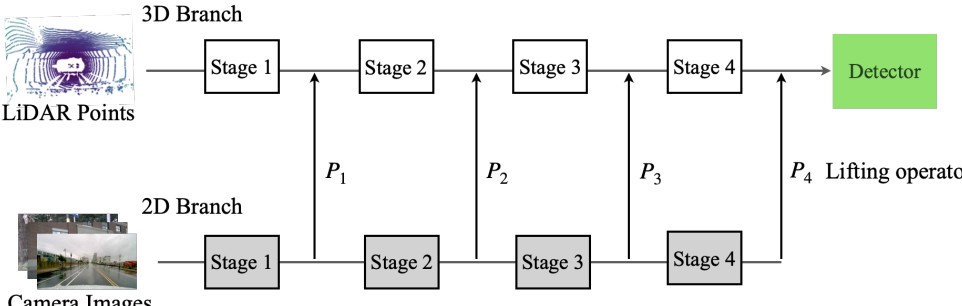

Figure 2: A generic 3D detection network with 2D feature fusion at different stages.

during fusion. However, the computation cost of attention scales linearly with the product of the number of voxels and the number of pixels, which significantly increases the model inference latency. For example, the attention module in Deepfusion (Li et al., 2022b) contributes to a more than 30% latency increase.

**Path and Cycle consistency**    To encourage the feature alignments in our work, we introduce a regularization term during training based on path consistency. Path and cycle consistency are common ways to validate whether samples in a graph can match after passing paths or cycles. In deep learning community, they are used in image matching (Zhou et al., 2015a; 2016; 2015b; Dwibedi et al., 2019), domain transfer and adaptation (Zhu et al., 2017; Hoffman et al., 2018; Yi et al., 2017) and regularizing the better feature quality in various of general tasks, including segmentation and detection (Zhang et al., 2019; Wang et al., 2013; 2014). For the first time, our work introduces path consistency into LiDAR and camera feature extraction and fusion to enable accurate 3D object detection.

## 3    BACKGROUND: 2D AND 3D FEATURE FUSION

The goal of Lidar-Camera feature fusion is to improve 3D detection tasks with additional dense 2D semantic information captured by cameras. A generic 3D detection network with multi-stage 2D feature fusion is shown in Figure 2. The network mainly consists of three components: 3D feature extraction branch, 2D feature extraction branch, and 2D-to-3D feature lifting.

**3D Feature Extraction**    Given a point cloud with a collection of $N$ points, i.e., $\{p_i\}_{i=1}^N$ ($p_i \in \mathbb{R}^3$), to detect objects of interest from the point cloud, a typical pre-processing step is to first apply voxelization on the points and transform the point cloud representation to a voxel volume representation. Denote $y_0 \in \mathbb{R}^{H \times W \times D}$ the transformed voxel volume, with $H, W, D$ representing the height, width, and depth in the 3D space, respectively. The 3D detection network is often built with a stack of a 3D feature backbone (e.g. Qi et al., 2017b; Zhou & Tuzel, 2018; Mao et al., 2021b) and a detector head (e.g. Lang et al., 2019; Shi et al., 2020; Yin et al., 2021a; Deng et al., 2021) with $y_0$ as the input. The 3D backbone often has multiple stages with down-sampling between each stage. Denote $y_i \in \mathbb{R}^{C_i \times H_i \times W_i \times D_i}$ the feature maps produced by stage $i$, where $C_i$ represents the channel size and $H_i, W_i, D_i$ denote the corresponding height, width and depth, respectively, after down-sampling.

**2D Feature Extraction**    Assume an RGB image, $x_0 \in \mathbb{R}^{3 \times H' \times W'}$, is captured along with the point cloud, with $H' \times W'$ as the image resolution. Similarly, with a modern feature backbone (e.g., ResNet50), one can extract 2D features for $x_0$ at different stages. Denote $x_i \in \mathbb{R}^{C_i' \times H_i' \times W_i'}$ the extracted 2D feature at stage $i$, where $C_i'$ is the channel size and $H_i' \times W_i'$ is the feature map size. The line of 2D and 3D feature fusion work aims at lifting 2D features $\{x_i\}$ to the 3D feature space and fusing the lifted features with 3D features $\{y_i\}$ to improve 3D detection tasks.

**2D-to-3D Feature Lifting**    One immediate challenge is that 2D and 3D feature maps have different spatial resolutions and relate to different locations in the real-world 3D space. Without loss of generality, consider fusing a 2D feature map $x_i$ from the stage $i$ in the 2D branch with a 3D feature

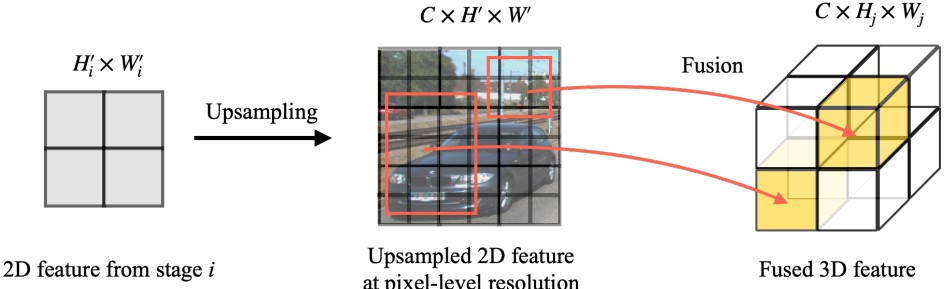

Figure 3: An illustration of lifting features from 2D to 3D at a deep stage. The upsampling is typically implemented with a feature pyramid network (Lin et al., 2017). Image is from (Geiger et al., 2013).

map $y_j$ from the stage $j$ in the 3D branch. Existing works (e.g., Chen et al., 2022a; Li et al., 2022b; Chen et al., 2022b) often first upsample $x_i$ to the input pixel-level resolution $H' \times W'$ using a feature pyramid network (Lin et al., 2017). Then for each voxel in the 3D volume $y_i$, one can use the camera parameters and query the corresponding 2D pixel-level feature for each voxel (e.g., Vora et al., 2020; Li et al., 2022b; Bai et al., 2022; Chen et al., 2022a). See Figure 3 for a demonstration. Note that each voxel might correspond to a region of a different size. In this work, we follow (Chen et al., 2022a) and take the feature vector at the center as the representation of the patch for simplicity. Finally, the fused 3D features could be a summation or concatenation between the lifted 2D features and original 3D features from the 3D backbone.

## 4 MOTIVATING EXAMPLE: CHALLENGES OF DEEP FUSION

Feature fusion at the input level is intuitive and straightforward, as each point from LiDAR can accurately locate its corresponding 2D pixels using the camera parameters. However, the accurate mapping between 2D and 3D is lost due to feature aggregation in both 2D and 3D backbone. Especially for deep layers, each voxel relates to a larger respective field, which has pixels that are far away from each other in the 3D space (e.g., car and background trees in Figure 3). Ideally, one would like each voxel and its corresponding 2D feature patch to encode the same part of the physical world. However, there is no guarantee of such correlation in the network since 2D and 3D features are both involved with non-linear transformations and down-sampling. Naively assigning each voxel with 2D features that are semantically unrelated might cause confusion of the 3D backbone and consequently degrade the 3D detection performance.

We verify our hypothesis in Figure 4 (a). We closely follow the setting in prior-art Focals Conv (Chen et al., 2022a), and alternate feature fusion from stage 1 to stage 4 as shown in Figure 2. In this case, we use VoxelNet (Zhou & Tuzel, 2018) with Focals Conv as the 3D backbone and ResNet-50 (He et al., 2016) as the 2D backbone. Both the 2D and 3D backbones have four stages. We report $\text{AP}_{3D}$ (R11) score on KITTI (Geiger et al., 2013). As shown in Figure 4 (a), the model trained with shallow feature fusion at stage 1 outperforms the baseline with no fusion, demonstrating the benefits of camera features for 3D detection. However, the performance drops consistently when fusing features at deeper layers due to potential semantic mis-alignment between 2D and 3D features. We propose a novel training regularization to improve deep feature fusion in the sequel.

## 5 METHOD: DEEP FUSION WITH PATH CONSISTENCY REGULARIZATION

We denote $x_i \sim y_i$ as an ideal semantic alignment between 2D features $x_i$ and 3D features $y_i$, in a way that there exists a lifting operator $P_i$, such that the lifted 2D features $\tilde{y}_i = P_i \circ x_i$ and $y_i$ are semantically aligned. We denote $\tilde{y}_i \approx y_i$ if $\tilde{y}_i$ and $y_i$ are semantically aligned, in a sense that $\tilde{y}_i$ and $y_i$ has a high similarity in the feature space.

For feature fusion between camera and LiDAR, $x_0 \sim y_0$ holds at the input layer where the mapping from 2D to 3D is exact, and $P_0$ can be constructed accurately with camera parameters. Assume $x_t \sim y_t$ hold for features at stage $t$, namely, there exists a $P_t$ such that $\tilde{y}_t = P_t \circ x_t \approx y_t$. It remains to show $x_{t+1} \sim y_{t+1}$ so that we can conclude features at all stages are semantically aligned from a mathematical induction point of view.

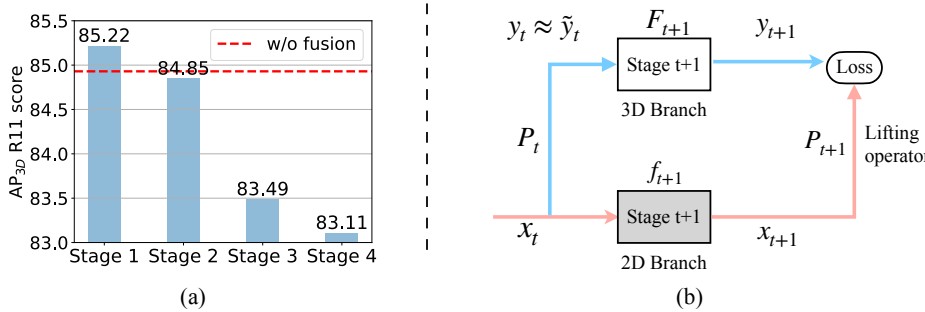

(a)                     (b)

Figure 4: (a) Illustration of performance degradation with naive deep feature fusion. Results are on the KITTI val split. The baseline setup without feature fusion achieves a 84.93% of $AP_{3D}(R11)$. (b) Illustration of out path-consistent loss.

In the following, we first show how $x_t$ and $y_t$ is transformed in the 2D branch and 3D branch, respectively. Then we present a path-consistent loss that encourages the network to learn semantically aligned 2D and 3D features, that is $x_{t+1} \sim y_{t+1}$ (or $P_{t+1} \circ x_{t+1} \approx y_{t+1}$).

**2D path** On the 2D branch, $x_t$ is first evolved with a stack of 2D convolutions layers (denoted as $f_{t+1}$) and then lifted by $P_{t+1}$. More precisely, the resulting 3D features can be written as the following,

$$y_{2D} = P_{t+1} \circ x_{t+1} = P_{t+1} \circ f_{t+1} \circ x_t, \quad (1)$$

where $x_{t+1}$ represents the output from $f_{t+1}$. Here we denote $A \circ B$ as applying operator $A$ on the output from $B$. See the red path in Figure 4 (b) for an illustration.

**3D path** The 3D counterpart $y_t$ is simply transformed by a number of 3D convolution layers (we denoted as $F_{t+1}$),

$$y_{3D} = F_{t+1} \circ y_t \approx F_{t+1} \circ \tilde{y}_t = F_{t+1} \circ P_t \circ x_t. \quad (2)$$

Here we replace $y_t \approx \tilde{y}_t = P_t \circ x_t$ by construction. See the blue path in Figure 4 (b) for an example.

**Path-consistent regularization** Our path-consistent regularization is simply defined as minimizing the distance between the lifted features $y_{2D}$ and the 3D features $y_{3D}$ such that $x_{t+1} \sim y_{t+1}$ as follows,

$$\mathcal{L}^t_{consistency} = loss(y_{2D}, \ y_{3D}) = loss(\underbrace{P_{t+1} \circ f_{t+1} \circ x_t}_{\text{2D path}}, \ \underbrace{F_{t+1} \circ P_t \circ x_t}_{\text{3D path}}), \quad (3)$$

where $loss(\cdot)$ denotes the loss function.

An alternative implementation of the consistency regularization is to minimize $loss(P_{t+1} \circ f_{t+1} \circ x_t, \ F_{t+1} \circ y_t)$. However, in practice, the 3D voxel feature $y_t$ is often much sparser compared to the 2D feature $x_t$. This loss is only defined where $y_t$ has values. Meanwhile, $F_{t+1} \circ y_t$ represents geometry-based features and $P_{t+1} \circ x_{t+1}$ represents color-based features, it is non-trivial to design an appropriate loss function to measure the distance before these two. While our consistency loss only depends on the same input $x_t$, the information from both paths is dense. Meanwhile, a simple loss function like negative cosine similarity or $\ell_1$ distance is sufficient since features are originated from the same color feature space.

**Path consistency for multi-stage networks** In a case of mutli-stage feature fusion wherein features are fused at both shallow and layers. Assume there are in total $n$ stages, we apply the path consistency defined in Eqn. 3 for all stages, that is,

$$\mathcal{L}_{consistency} = \sum_{t=1}^{n-1} \mathcal{L}^t_{consistency} = \sum_{t=1}^{n-1} loss(\underbrace{P_{t+1} \circ f_{t+1} \circ x_t}_{\text{2D path}}, \ \underbrace{F_{t+1} \circ P_t \circ x_t}_{\text{3D path}}). \quad (4)$$

We refer to Eqn 4 as our path consistency loss.

**Algorithm** We augment the standard 3D detection loss with our path consistency loss. Overall, our training objective can be formulated as follows,

$$\mathcal{L} = \mathcal{L}_{3D} + \alpha\mathcal{L}_{\text{consistency}}. \tag{5}$$

Here $\mathcal{L}_{3D}$ is the main loss for the 3D task that affects all the parameters, $\mathcal{L}_{\text{consistency}}$ is our path-consistent regularization, and $\alpha$ is a hyper-parameter that controls the magnitude of the regularization. As our goal is to regularize the 2D branch to improve the 3D detection task, we only back-propagate our path consistency loss through the lifting operators and the 2D backbone while stopping the corresponding gradients on the 3D backbone.

## 6    EXPERIMENTS

We evaluate our method, PathFusion, on both KITTI (Geiger et al., 2013) and nuScenes (Caesar et al., 2020). We choose Focals Conv (Chen et al., 2022a) as our base model. We add our path-consistent loss on top of Focals Conv training to facilitate feature fusion between the 2D and 3D branches. In alignment with Focals Conv, we found that naively fusing LiDAR and camera features at deeper layers leads to significant performance degradation (see e.g., Figure 4 (a)). However, our path-consistent regularization learns semantically aligned transforms from 2D to 3D, yielding consistent improvements as more deep features are fused. Specifically, comparing to Focals Conv with the same detector Voxel-RCNN, we improved the AP$_{3D}$ (R11) from 85.22% to 85.84% on KITTI and improved mAP from 70.1% to 71.3% on nuScenes testing split, respectively.

**PathFusion settings** Focals Conv fuses stage 1 features from the 2D branch. We denote this implementation as Focals Conv-F. On top of the Focals Conv-F setting, we add deep-level feature fusion with our path-consistent regularization. As defined in Eqn. 4, we do not apply the path consistency loss on stage 1, as the misalignment issue is only significant at deeper layers.

We choose cosine as our consistency loss and set $\alpha = 0.01$ in Eqn. 5 as default unless otherwise specified. Additionally, our consistency regularization only affects parameters stage-wise. For example, our consistency loss computed at the end of stage $i$ only back-propagates to parameters in stage $i$ and the corresponding lifting operator $P_i$.

### 6.1    RESULTS ON KITTI

**Dataset** We follow the setting on KITTI (Geiger et al., 2013) by splitting the dataset into 3,717 for training and 3,769 for validation. Each KITTI example contains one LiDAR input together with two camera inputs. We only use the LiDAR input and one RGB from the left camera for a fair comparison with Focals Conv (Chen et al., 2022a).

**Settings** Following the setup in Focals Conv (Chen et al., 2022a), we use Voxel RCNN (Deng et al., 2021) as as the overall detection framework, in which we use VoxelNet (Zhou & Tuzel, 2018) with Focals Conv as the 3D backbone and ResNet-50 (He et al., 2016) as the 2D backbone. Following Focals Conv, the ResNet-50 is pretrained on COCO (Lin et al., 2014) using Deeplabv3 (Chen et al., 2017). To enable deep-level fusion, we introduce feature fusion in stage 4 and keep the original shallow-level fusion in stage 1.

We strictly follow the training setting used in Focals Conv-F and train the model for 80 epochs with batch size of 16, and we use Adam optimizer with a learning rate 0.01. We report the mean Average Precision (mAP) metric. The mAP calculates the 3D bounding box recall score on 11 positions. The metric is referred to as AP$_{3D}$(R11).

**Results** We summarize our results in Table 1. We report the AP$_{3D}$(R11) score evaluated in easy, moderate and hard levels following the previous works (Chen et al., 2022a; Deng et al., 2021; Shi et al., 2020). As we can see from Table 1, our method outperforms Focals Conv-F by 0.41%, 0.62%, 0.12% in three levels and achieves 90.23%, 85.84% and 85.31% on AP$_{3D}$(R11). These results suggest that our proposed PathFusion learns better feature fusion and thus provides consistent improvement.

| Method | Modality | Easy | Moderate | Hard |
|---|---|---|---|---|
| PointPillars (Lang et al., 2019) | L | 86.62 | 76.06 | 68.91 |
| SECOND (Yan et al., 2018) | L | 88.61 | 78.62 | 77.22 |
| Point R-CNN (Shi et al., 2019) | L | 88.88 | 78.63 | 77.38 |
| Part-$A^2$ (Shi et al., 2021) | L | 89.47 | 79.47 | 78.54 |
| 3DSSD (Yang et al., 2020) | L | 89.71 | 79.45 | 78.67 |
| STD (Yang et al., 2019) | L | 89.70 | 79.80 | 79.30 |
| SA-SSD (He et al., 2020) | L | 90.15 | 79.91 | 78.78 |
| PV-RCNN (Shi et al., 2020) | L | 89.35 | 83.69 | 78.70 |
| VoTr-TSD (Mao et al., 2021b) | L | 89.04 | 84.04 | 78.68 |
| Pyramid-PV (Mao et al., 2021a) | L | 89.37 | 84.38 | 78.84 |
| Voxel R-CNN (Deng et al., 2021) | L | 89.41 | 84.54 | 78.93 |
| Focals Conv (Chen et al., 2022a) | L | 89.52 | 84.93 | 79.18 |
| LargeKernel3D (Chen et al., 2022b) | L | 89.52 | 85.07 | 79.32 |
| F-PointNet (Qi et al., 2018) | LC | 83.76 | 70.92 | 63.65 |
| PointSIFT+SENet (Zhao et al., 2019) | LC | 85.62 | 72.05 | 64.19 |
| 3D-CVF (Yoo et al., 2020) | LC | 89.67 | 79.88 | 78.47 |
| Focals Conv-F (Chen et al., 2022a) | LC | 89.82 | 85.22 | 85.19 |
| *PathFusion (ours)* | LC | **90.23** | **85.84** | **85.31** |

Table 1: Comparison on KITTI. Results are AP$_{3D}$ (R11) for Car on the val split. L represents LiDAR input and LC represents both LiDAR and camera input. Focals Conv-F denotes the implementation with shallow fusion on stage 1 in Figure 2.

## 6.2 RESULTS ON NUSCENES

**Dataset** nuScenes (Caesar et al., 2020) [1] is a large-scale 3D object detection dataset for self-driving. It consists of 700 scenes for training, 150 scenes for validation and 150 scenes for testing. Each scene has a sequence of frames and each frame contains data from one LiDAR and 6 cameras. In total, nuScenes contains 1.4M camera images and 390K LiDAR sweeps.

**Settings** Following Focals Conv-F (Chen et al., 2022a), we use VoxelNet as our 3D backbone and use ResNet-50 pretrained using DeeplabV3 (Chen et al., 2017) on COCO (Lin et al., 2014) as the 2D backbone. We train the model for 20 epochs with batch size 32 and Adam Optimizer with 0.001 learning rate. We apply a cosine scheduler to decrease the learning rate to $1e^{-4}$. For the data augmentation, same as Focals Conv-F, we apply rotation, flip, translate, rescale and ground truth sampling augmentation during training. We extend the Focals Conv-F to fuse 2D and 3D features at both stage 1 and stage 2 (see Figure 2).

**Results** We evaluate on both the nuScenes validation and test set, and report our results in Table 2 and Table 3, respectively. On the validation split, we achieve a 1.5% improvement on mAP and 0.7% improvement on NDS (NuScenes Detection Score), respectively, compared to Focals Conv-F.

We further evaluate our model on the nuScenes test server. Following Liu et al. (2022c); Liang et al. (2022); Chen et al. (2022a), we report results with and without testing time data augmentation. For the testing-time augmentations, we follow Chen et al. (2022a) and use double flip and rotation with yaw angles in $[-6.25°, 0°, 6.25°]$. We observe a consistent improvements on Focals Conv across all the settings by regularizing the training with our path-consistent loss. Specifically, our best result outperforms Focals Conv-F by 1.2%, 1.3%, 1.2% mAP and 0.8%, 0.8%, 0.7% NDS on the test set for the settings without and with testing-time data augmentation.

## 6.3 ABLATION STUDIES

**Choices of fusion configurations** To study the importance of feature fusion at deep layers, we alternate the feature fusion at different stages.

---
[1]https://www.nuscenes.org/

| Method | mAP | NDS | Car | Truck | Bus | Trailer | C.V. | Ped | Mot | Byc | T.C. | Bar |
|---|---|---|---|---|---|---|---|---|---|---|---|---|
| Focals Conv-F | 63.8 | 69.4 | 86.5 | 58.5 | 72.4 | 41.2 | 23.9 | 86.0 | 69.0 | 55.2 | 76.8 | 69.1 |
| *PathFusion (ours)* | **65.3** | **70.1** | 86.8 | 61.4 | 72.1 | 42.3 | 26.6 | 87.0 | 75.2 | 61.0 | 77.5 | 66.2 |

Table 2: Comparison with other methods on nuScenes val split *without* testing-time augmentations.

| Method | Modality | mAP | NDS | Car | Truck | Bus | Trailer | C.V. | Ped | Mot | Byc | T.C. | Bar |
|---|---|---|---|---|---|---|---|---|---|---|---|---|---|
| PointPillars (Lang et al., 2019) | L | 30.5 | 45.3 | 68.4 | 23.0 | 28.2 | 23.4 | 4.1 | 59.7 | 27.4 | 1.1 | 30.8 | 38.9 |
| 3DSSD (Yang et al., 2020) | L | 42.6 | 56.4 | 81.2 | 47.2 | 61.4 | 30.5 | 12.6 | 70.2 | 36.0 | 8.6 | 31.1 | 47.9 |
| CBGS (Zhu et al., 2019) | L | 52.8 | 63.3 | 81.1 | 48.5 | 54.9 | 42.9 | 10.5 | 80.1 | 51.5 | 22.3 | 70.9 | 65.7 |
| HotSpotNet (Chen et al., 2020b) | L | 59.3 | 66.0 | 83.1 | 50.9 | 56.4 | 53.3 | 23.0 | 81.3 | 63.5 | 36.6 | 73.0 | 71.6 |
| CVCNET (Chen et al., 2020a) | L | 58.2 | 66.6 | 82.6 | 49.5 | 59.4 | 51.1 | 16.2 | 83.0 | 61.8 | 38.8 | 69.7 | 69.7 |
| CenterPoint (Yin et al., 2021a) | L | 58.0 | 65.5 | 84.6 | 51.0 | 60.2 | 53.2 | 17.5 | 83.4 | 53.7 | 28.7 | 76.7 | 70.9 |
| CenterPoint[†] | L | 60.3 | 67.3 | 85.2 | 53.5 | 63.6 | 56.0 | 20.0 | 84.6 | 59.5 | 30.7 | 78.4 | 71.1 |
| Focals Conv (Chen et al., 2022a) | L | 63.8 | 70.0 | 86.7 | 56.3 | 67.7 | 59.5 | 23.8 | 87.5 | 64.5 | 36.3 | 81.4 | 74.1 |
| LargeKernel3D (Chen et al., 2022b) | L | 65.3 | 70.5 | 85.9 | 55.3 | 66.2 | 60.2 | 26.8 | 85.6 | 72.5 | 46.6 | 80.0 | 74.3 |
| PointPainting (Vora et al., 2020) | LC | 46.4 | 58.1 | 77.9 | 35.8 | 36.2 | 37.3 | 15.8 | 73.3 | 41.5 | 24.1 | 62.4 | 60.2 |
| 3DCVF (Yoo et al., 2020) | LC | 52.7 | 62.3 | 83.0 | 45.0 | 48.8 | 49.6 | 15.9 | 74.2 | 51.2 | 30.4 | 62.9 | 65.9 |
| FusionPainting (Xu et al., 2021b) | LC | 66.3 | 70.4 | 86.3 | 58.5 | 66.8 | 59.4 | 27.7 | 87.5 | 71.2 | 51.7 | 84.2 | 70.2 |
| MVF (Yin et al., 2021c) | LC | 66.4 | 70.5 | 86.8 | 58.5 | 67.4 | 57.3 | 26.1 | 89.1 | 70.0 | 49.3 | 85.0 | 74.8 |
| PointAugmenting (Wang et al., 2021a) | LC | 66.8 | 71.0 | 87.5 | 57.3 | 65.2 | 60.7 | 28.0 | 87.9 | 74.3 | 50.9 | 83.6 | 72.6 |
| Focals Conv-F (Chen et al., 2022a) | LC | 67.8 | 71.8 | 86.5 | 57.5 | 68.7 | 60.6 | 31.2 | 87.3 | 76.4 | 52.5 | 84.6 | 72.3 |
| *PathFusion (ours)* | LC | **69.0** | **72.6** | 87.5 | 59.8 | 69.3 | 62.0 | 34.7 | 87.6 | 77.3 | 53.7 | 85.3 | 72.9 |
| Focals Conv-F[†] | LC | 68.9 | 72.8 | 86.9 | 59.3 | 68.7 | 62.5 | 32.8 | 87.8 | 78.5 | 53.9 | 85.5 | 72.8 |
| *PathFusion (ours)*[†] | LC | **70.2** | **73.6** | 88.0 | 61.4 | 69.4 | 64.1 | 34.9 | 88.5 | 80.2 | 54.8 | 85.9 | 73.2 |
| Focals Conv-F[‡] | LC | 70.1 | 73.6 | 87.5 | 60.0 | 69.9 | 64.0 | 32.6 | 89.0 | 81.1 | 59.2 | 85.5 | 71.8 |
| *PathFusion (ours)* [‡] | LC | **71.3** | **74.3** | 88.7 | 62.6 | 70.1 | 64.9 | 35.9 | 89.8 | 82.5 | 59.6 | 86.0 | 72.4 |

Table 3: Comparison on the nuScenes test set. L represents LiDAR input and LC represents both LiDAR and camera input. [†] denotes double flip and [‡] denotes double flip and rotation testing-time augmentation, respectively.

We first evaluate on the KITTI dataset. As we can see from Table 4, without our path-consistent regularization, the best detection result is achieved when only the feature from stage 1 is fused. In contrast, our method leads to consistent improvements. And our best results are achieved by fusing all the stages. Meanwhile, fusing both shallow features from stage 1 and deeper features from stage 4 leads to a similar performance compared to the result from fusing all the stages.

We further evaluate on nuScenes. Here, to reduce the experiment turnaround time, we follow Focals Conv (Chen et al., 2022a) and only train on a $\frac{1}{4}$ split of the full training set. As we can see from Table 5, without our path-consistent regularization, both the mAP and NDS scores decrease significantly when deep features are fused. This trend is in consistent with what we observed on KITTI. However, our method learns better feature fusion and achieves 0.7% improvements with path-consistent regularization.

| Method | w/o Fusion | Stage 1 | Stage 2 | Stage 3 | Stage 4 | Stage 1&2 | Stage 1&3 | Stage 1&4 | All Stages |
|---|---|---|---|---|---|---|---|---|---|
| w/o path consistency | 84.93 | 85.22 | 84.85 | 83.49 | 83.11 | 85.01 | 84.85 | 84.02 | 83.97 |
| w/ path consistency | - | - | **85.69** | **85.76** | **85.78** | **85.79** | **85.82** | **85.84** | **85.88** |
| *Relative improvements* | - | - | 0.84 | 2.27 | 2.67 | 0.78 | 0.97 | 1.82 | 1.91 |

Table 4: Ablation Study on different fusion configurations. Results are $AP_{3D}(R11)$ on the KITTI validation set. All Stages represent fusing Stage 1-4.

**Robustness of our method** We found our method leads to significantly improved training stability. Specifically, we use the official Focals Conv-F repo [2] and repeat the experiments 10 times with different random seeds and report the $AP_{3D}(R11)$ at the moderate level. We report our findings in Table 6. Compared with Focals Conv-F, the results of our method has a small standard deviation

---

[2]https://github.com/dvlab-research/FocalsConv

|  | mAP | NDS |
|---|---|---|
| Shallow-level fusion | 61.7 | 67.2 |
| Deep-level fusion | 60.7 | 65.6 |
| Shallow & Deep level fusion | 61.5 | 67.0 |
| + path consistency | **62.4** | **67.8** |

Table 5: Improvement over multi-modal baseline trained on nuScenes $\frac{1}{4}$ and evaluate on val set.

(i.e., 0.13), our worst run also outperfroms the best run of Focal Convs-F. We report the average performance of our method in Table 1.

| Method | Mean | Min | Max |
|---|---|---|---|
| Focals Conv-F | $85.06 \pm 0.34$ | 84.59 | 85.43 |
| PathFusion (ours) | $85.84 \pm 0.13$ | 85.70 | 86.07 |

Table 6: Comparison on KITTI with 10 random trials. Results are $AP_{3D}$ (R11) at the moderate level. *Min* and *Max* denotes the minimum and maximum $AP_{3D}$ (R11) achieved out of the 10 runs.

**Impact of gradient stopping on 3D branch**    Gradient stopping is an important trick to regularize the corresponding parameter part of path consistency loss properly. The reason is two-fold: 1) without gradient stopping, the path-consistent alone has trivial solutions by producing zero activations for both the 2D and 3D branches. 2) the task of interest is 3D detection; hence, it is intuitive only to regularize the 2D branch to facilitate the 3D task without interfering with the 3D branch. To verify, we remove the gradient stopping on the 3D backbone and evaluate on the nuScenes $\frac{1}{4}$ split. We found the mAP and NDS drop significantly from 62.4%, 67.8% to 56.7% and 63.0%, respectively, validating the effectiveness of gradient stopping on the 3D branch.

**Consistency loss design**    We experiment with the cosine and $\ell_1$ loss as our consistency loss function. And in the meantime, we study the impact of our consistency coefficient $\alpha$ and test on the 1/4 nuScenes split in Table 7. As we can see in Table 7, our method outperforms the baseline (mAP 61.7% and NDS 67.2%) in all settings. And the choice of cosine loss with a coefficient of $\alpha = 0.01$ provides the best performance.

| Loss Type | Loss weight ($\alpha$) | mAP | NDS |
|---|---|---|---|
|  | 0.1 | 62.1 | 67.5 |
| Cosine | 0.01 | **62.4** | **67.8** |
|  | 0.001 | 62.3 | 67.7 |
|  | 0.1 | 61.8 | 67.3 |
| $\ell_1$ | 0.01 | 62.0 | 67.5 |
|  | 0.001 | 62.0 | 67.4 |

Table 7: Ablation study on the impact of different path-consistent loss design and impact of the loss coefficient $\alpha$. The result is trained on nuScenes $\frac{1}{4}$ and evaluate on val set.

## 7 CONCLUSION

In this work, we proposed a path consistency loss to improve deep fusion between LiDAR features and camera features. Our method works by encouraging the 2D branch to follow the transformations learned in the 3D branch and hence, producing complementary information that is semantically aligned to the 3D features. We applied our method to improve prior-art Focals Conv, and our method leads to significant improvements on both KITTI and nuScenes datasets. Specifically, our PathFusion achieves 71.3% mAP on the nuScenes test set, 1.2% better than the result from Focals Conv-F.

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
