# OpenReview forum: "PathFusion: Path-consistent Lidar-Camera Deep Feature Fusion"
_ICLR.cc/2023/Conference — Submitted to ICLR 2023_

### Official Review · Reviewer_gP9m · 2022-10-22

**Confidence:** 4
**Correctness:** 3
**Technical Novelty And Significance:** 3
**Empirical Novelty And Significance:** 2
**Recommendation:** 5

**Clarity, Quality, Novelty And Reproducibility:**

As for the clarity and quality of the text itself, the authors initially state that they build on FocalsConv and introduce an additional loss function for path consistency. However, from section two and onwards, the separation between proposed novelties and existing methods is far from obvious. I personally find that this is an important weakness in the current version of the manuscript and something that I hope the authors can improve substantially; it should be easy to fix.

Another important aspect is that I don't think the description of the state-of-the-art is somewhat misleading in the sense they argue that most methods are shallow, whereas many modern methods are arguably deep according to the terminology used in the paper.

**Strength And Weaknesses:**

The proposed idea (the loss function) is interesting, and I believe it has some merit. The fact that the loss function improves performance on both KITTI and nuScenes data is, of course, a significant strength. At the same time, it is also a simple idea, and I don’t think the authors have done enough to motivate that it is important.

A few aspects that would be good to discuss further:

* How general is this idea? The authors only demonstrate it on a very specific architecture called FocalsConv, and I personally doubt that FocalsConv will be commonly used in the future. Can the authors mention other networks that the loss function could be combined with to motivate that the generality of the idea goes beyond FocalsConv. Ideally, the authors should include such examples in the evaluation, but I am not that is necessary.

* What is the relation between the proposed loss and self-supervised learning? My understanding is that the loss function can be used in a self-supervised manner. In the current version, this has not been discussed or explored in a satisfying manner.

**Summary Of The Paper:**

The submitted manuscript considers the task of fusing camera and lidar data to produce 3D object detections. The paper focuses on a specific type of fusion mechanism where 2D camera features are mapped to 3D for inference and discusses the pros and cons of late vs. early fusion. The main contribution of the paper is a loss function that penalizes inconsistencies across different paths through the network. The experiments demonstrate that the proposed loss function can lead to improved performance on both the KTTI and NuScenes benchmarks.

**Summary Of The Review:**

To conclude,

*  (pros) the proposed loss function is intuitively appealing,
*  (pros) the results demonstrate that the loss function improves performance,
*  (con) the presented solution is not state-of-the-art,
*  (con) some aspects are not sufficiently well explored,
* (con) the paper could be more well written.

My current recommendation is marginally below acceptance, but I would be willing to change that if the paper is adjusted.

---

> ### Author Response · Authors · 2022-11-17
> **Offical Response to Reviewer gP9m**
>
> We sincerely thank Reviewer gP9m for your thoughtful feedback.
>
> __Q1: The proposed idea (the loss function) is interesting, and I believe it has some merit. The fact that the loss function improves performance on both KITTI and nuScenes data is, of course, a significant strength. At the same time, it is also a simple idea, and I don’t think the authors have done enough to motivate that it is important.__
>
> __A__: We thank Reviewer gP9m for your recognition on the simplicity of our method and the significance of our empirical results.
> We also believe the merit of our proposed method is on its simplicity and effectiveness.
> Nevertheless, its simplicity, our solution is less than obvious. The majority of existing works often focus on
> 1)either architecture designs (e.g., Deep-interaction, TransFusion, DeepFusion) 2) or new surrogate tasks (e.g., BEVFusion).
> Instead, our work offers a new perspective from training regularization,
> the direction, which is largely overlooked by the community.
> We optimistically believe our work would motivate more future research in this direction.
>
> Our motivation is mostly emphasized in our introduction, section 4 and section 6. Specifically, we would like to refer Reviewer gP9m to our ablation study at Table 4 and Table 5 for a specific example as our motivation.
>
> In table 4 at row _w/o path consistency_, we show that naively fusing deep layer features hurts the performance.
> In table 5 at row _Shallow & Deep level fusion_, we show a similar performance degradation by naively fusing deep features on nuScenes.
> All these results confirm the importance of a better feature fusion solution. Our method provides a simple solution from the training perspective, that is to regularize consistency between features from two different paths without the need of change the architecture.
>
> In this way, our method doesn't lead to any computational overhead at inference time while improving the 3D detection significantly by learning better representations in the network.
>
> __Q2: How general is this idea?.__
>
> __A__: The reason we chose FocalsConv as our base model is due to its superior performance.
> By the time of submission, FocalsConv yields 70.1 mAP on the nuScenes leaderboard. FocalsConv achieves the best performance among all published works. Other solutions like BEVFusion with better numbers are either 1) not fully open-sourced  2) or still under peer review.
> We carefully choose the best possible baseline in the area to compare with to demonstrate the effectiveness of our method.
>
> In fact, FocalsConv is built with VoxelRCNN for KITTI and CenterPoint for nuScense, which are two very common detectors and wildly used in many models, e.g., FusionPainting, PointAugmenting, MVP. The underlying backbone is VoxelNet, which is also a common encoder for 3D backbone. FocalConvs mainly propose an additional layer inside the VoxelNet to enhance the 3D representation. So our framework actually can easily extend to other methods that use VoxelNet as the 3D encoder or use CenterPoint or VoxelRCNN as the detector.
>
> __Q3: What is the relation between the proposed loss and self-supervised learning? My understanding is that the loss function can be used in a self-supervised manner. In the current version, this has not been discussed or explored in a satisfying manner.__
>
> __A__: In this work, we focused on the supervised learning setting and aimed to improve 3D detection performance.
> Thank Reviewer gP9m for your insightful suggestion. Indeed, our path consistency might also be used to pre-train the lifting operators as well as the 3D backbone. We will consider the extension of our work to self-supervised learning a future step. Thank you!
>
> __Q4: As for the clarity and quality of the text itself, the authors initially state that they build on FocalsConv and introduce an additional loss function for path consistency. However, from section two and onwards, the separation between proposed novelties and existing methods is far from obvious. I personally find that this is an important weakness in the current version of the manuscript and something that I hope the authors can improve substantially; it should be easy to fix.__
>
> __A__: Thank you for your suggestion. We've updated our draft accordingly.
>
> __Q5: Another important aspect is that I don't think the description of the state-of-the-art is somewhat misleading in the sense they argue that most methods are shallow, whereas many modern methods are arguably deep according to the terminology used in the paper.__
>
> __A__: Most of the existing work only fuse shallow / deep 2D layers to shallow 3D layers.
>
> As observed in the literature, see e.g., PointPaiting, PonintAugment, DeepFusion, as well as verified empirically in Figure 4(a), Table 4, and Table 5 in this work, naively fusing deep 2D features and deep 3D features is problematic. This is the challenge we aim to solve in this work.

---

> > ### Author Response · Authors · 2022-11-17
> > **Offical Response to Reviewer gP9m - Part 2**
> >
> > __Q6: The presented solution is not state-of-the-art.__
> >
> > __A__: We emphasize our results are indeed state-of-the-art at the time of submission compared with ONLY published paper. Indeed, nuScenes is a very active benchmark and there are a few new methods on the leaderboard that show better results than ours as of today with better 3D branch architecture design, powerful pretrained 2D branches with more data like nuImage and stronger 2D branch model like Swin Transformer. We believe our method is orthogonal to this add-on and can get better performance by adapting these new training technologies.

---

### Official Review · Reviewer_W3vh · 2022-10-25

**Confidence:** 3
**Correctness:** 4
**Technical Novelty And Significance:** 2
**Empirical Novelty And Significance:** 2
**Recommendation:** 5

**Clarity, Quality, Novelty And Reproducibility:**

This paper is generally well-written and I believe it's easy to implement.

A few questions to better understand this paper:
1) In figure 4(b),  should the input of F_{t+1} be concatenation or summation of y_t and P_t\circle x_t because the 3D branch is handling fused features? If so, how to separate lifted 2D features from this 3D branch? And are feature channel size the same across different stages?
2) Is data augmentation applied in KITTI experiments? How does the proposed method handle misalignment caused by data augmentation especially gt sampling?

A minor question: in Section 6 paragraph PathFusion settings 'back-progates to parameters in stage i and the corresponding lifting operator P_i'. The lifting operator is upsampling and mapping the center of 2D regions to 3D voxels. There should be no parameters in lifting operators?

**Strength And Weaknesses:**

Strength:
The approach is reasonable. Paper is generally well-written. Ablation study is extensive.
Weaknesses:
- The biggest concern is the results only show marginal gain compared to baseline. Especially on KITTI the results are known to be volatile.
- This paper claims other fusion methods are not optimized in model complexity but no latency comparison is provided.

**Summary Of The Paper:**

Empirical results show deep 3d fusion of lidar and camera has inferior performance compared to early stage fusion. A possible reason is the misalignment in deep fusion when deep features have large receptive fields and include overlapping regions of large areas. This paper aims to enable path-consistency between early features and deep features to enforce alignment in deep features. Path consistency is negative cosine similarity or l1 distance between features of lifted 2D features and 3D features whose input is lifted 2D features from previous stage . The experiments show 1.2%+ mAP on nuscenes test set and 0.6%+ on KITTI AP3d moderate compared to its baseline Focals Conv.

**Summary Of The Review:**

The biggest concern is the results only show marginal gain compared to baseline. Without latency comparison to other fusion methods and other sota lidar-only methods, the proposed method has not justify the potential of wide application.

---

> ### Author Response · Authors · 2022-11-17
> **Official Response to Reviewer W3vh**
>
> We sincerely thank Reviewer W3vh for your thoughtful feedback.
>
> __Q1: The biggest concern is the results only show a marginal gain compared to the baseline. Especially on KITTI the results are known to be volatile.__
>
> __A__: __Improvements on KITTI__: We would like to emphasize that our results on KITTI are significant for the following reasons.
>
> First, we are using the same architecture as FocalsConv. Thus the performance gain is purely brought by the regularization of the feature fusion.
>
> Then, we agree with Reviewer W3vh that the round-to-round variation for prior works (e.g., Focals Conv-F) is large.
> We also confirmed this issue in our ablation study. See paragraph _Robustness of our method_ and Table 6.
>
> In our results, we conduct 10 random trials for each method.
> In Table 6, we show that the results of Focals Conv-F have a big variance and the standard deviation is 0.34. However, our method is more robust and consistent. Even our worst trial outperforms the best trial of FocalsConv-F. In table 1, we report the best performance for Focals Conv-F to be consistent with that in the Focals Conv paper, while we report the average performance for our method.
>
> Moreover, as shown in Table 1, the relative improvement between existing papers (after 2021) is often around 0.5% on AP3D moderate. In contrast, our result is 0.8% better compared with Focals Conv-F (see Table 6). Hence, we believe this is a meaningful improvement.
>
> __Improvements on nuScenes__: Considering the most popular mAP metric, as shown in Table 3, the relative improvements between papers published after 2021 (e.g., FusionPainting, MVF, PointAugmenting and Focals Conv-F) are often around 1.0%. Our method improves Focals Conv-F by more than 1.2% in various settings, demonstrating the significance of our results to the community standard. Meanwhile, by the time of submission, our method ranks first on nuScenes when compared with only published papers.
>
> __Q2: This paper claims other fusion methods are not optimized in model complexity but no latency comparison is provided.__
>
> __A__: Latency comparison is not the focus of the paper. The claim on model
> complexity is straightforward because: our method works as a regularization
> that does not impact the inference time; in contrast, methods, e.g.,
> DeepFusion, adds extra attention modules between the 2D and 3D branches to alleviate the semantic misalignment issue. The complexity of attention modules increases linearly w.r.t the product of a number of voxels and 2D resolutions and often leads to a ~2X slow down at inference time based on our re-implement of the attention modules proposed in DeepFuison.
>
> __Q3: In figure 4(b), should the input of__ $F_{t+1}$ __be a concatenation or summation of__ $y_t$ __and__ $P_t\circ x_t$ __because the 3D branch is handling fused features? If so, how to separate lifted 2D features from this 3D branch? And are feature channel size the same across different stages?__
>
> __A__: Concatenation or summation is only applied during the actual fusion operation. As we illustrated in Equation (3) and (4), our path consistency loss only depends on $x_t$.
>
> __Q4: Is data augmentation applied in KITTI experiments? How does the proposed method handle misalignment caused by data augmentation, especially gt sampling?__
>
> __A__: We follow the FocalsConv setup and use gt sampling as well as re-scale, rotation, and translation for the data augmentation in KITTI and nuScenes.
> For the spatial transformation misalignment, we follow FocalsConv to reverse the coordinates of sparse features with the recorded transformation parameters.
> For gt-sampling,
> we follow FocalsConv that first trains the model with gt-sampling and then drops gt-sampling at the last four epochs to alleviate the misalignment issue caused by  gt-sampling.
>
> __Q5: A minor question: in Section 6 paragraph PathFusion settings 'back-progates to parameters in stage i and the corresponding lifting operator__ $P_i'$. __The lifting operator is upsampling and mapping the center of 2D regions to 3D voxels. There should be no parameters in lifting operators?__
>
> __A__: In this work, we follow the design in existing solutions like FocalsConv/ DeepFusion and always upsample the 2D features to a fixed resolution (e.g., the original resolution ) with a learnable FPN. The FPN is considered as part of the lifting operator. Please see Figure 3 for an illustration.

---

> > ### Author Response · Authors · 2022-11-17
> > **Official Response to Reviewer W3vh - Part 2**
> >
> > __Q6: Without latency comparison to other fusion methods and other sota lidar-only methods, the proposed method has not justified the potential of wide application.__
> >
> > __A__: Our method's main focus is providing an easy design for enabling deep-level fusion and multi-level fusion. Latency is not our main claim. For the comparison with the fusion method, we already discussed in Q2. For the comparison with the LiDAR-only method, we refer reviewers to take DeepFusion [1] Figure 8 as an example. They scale up the LiDAR-only method model size to compete with the Fusion-based method under the same latency to show the power of the fusion method. Since scaling up the LiDAR-only model can only serve as an ablation study since it is not the best hyperparameter setup, we do not include this comparison in our paper. We stress our boarderly use by showing the simple loss function that does not introduce any new module and the orthogonal
> > design with other lines of new architectures. So it has a great potential to adapt to various future works.
> >
> > [1] Yingwei Li, Adams Wei Yu, Tianjian Meng, Ben Caine, Jiquan Ngiam, Daiyi Peng, Junyang Shen,
> > Yifeng Lu, Denny Zhou, Quoc V Le, et al. Deepfusion: Lidar-camera deep fusion for multi-modal
> > 3d object detection. In Proceedings of the IEEE/CVF Conference on Computer Vision and Pattern
> > Recognition, pp. 17182–17191, 2022b

---

### Official Review · Reviewer_3wCP · 2022-10-27

**Confidence:** 4
**Clarity, Quality, Novelty And Reproducibility:** Please see the strength and weaknesses.
**Correctness:** 3
**Technical Novelty And Significance:** 2
**Empirical Novelty And Significance:** 2
**Recommendation:** 5

**Strength And Weaknesses:**

Strength:
+ The motivation is good. Compared to the previous methods that only fuse the features in shallow layers, this method considers the fusion of deep-layer features which is valuable to multi-modal detection.
+ The method is easy to follow and reproducable. The definition of path consistency loss is simple.
+ Experiments on KITTI and NuScenes datasets demonstrate the robustness of the proposed method.

Weaknesses:
- The contribution is slightly incremental. The main part of this work is the design of path consistency regularization.
- The forced alignment of the camera and LiDAR features is doubtful. As mentioned in the introduction, the LiDAR branch mainly extracts the geometry features, while the image branch produces the semantic features. Though the 2D semantic features are transformed into the 3D space, the distribution of lifted features remains intrinsic differences. Instead of aligning the features, learning an accurate transformation is more critical.
- The description of the lift operator is unclear. Are the 2D features in different stages upsampled to the origin resolution before the lift transformation? Is that possible to adjust calibration parameters to directly lift 2D features to 3D features?
- In practice, the point cloud and image cannot be aligned perfectly due to the synchronization error and motion distortion. For these cases, the shallow fusion is not reliable. Is deep fusion helpful in such situations?

**Summary Of The Paper:**

This paper proposes a new technique called PathFusion that aims to fuse the LiDAR and camera features, especially the deep layer features. In order to fuse the features at multiple stages, the method design a path-consistency loss to lift and align the 2D features in the camera branch to the 3D features in the LiDAR branch. With the path-consistent regularization, the PathFusion further improves the detection accuracy on both KITTI and NuScenes datasets compared to the baseline Focals Conv method.

**Summary Of The Review:**

This paper present PathFusion to fuse the multi-stage features between LiDAR and camera branches to improve 3D detection. A path consistency regularization is designed to align the 2D image and 3D lidar features. Experiments validate the performance of the proposed method. However, the contribution is limited, and the practice of forced alignment is doubtful. Considering the above factors, I vote for the weak reject in this round of review.

---

> ### Author Response · Authors · 2022-11-17
> **Offical Response to Reviewer 3wCP**
>
> We sincerely thank Reviewer 3wCP for your thoughtful feedback.
>
> __Q1: The contribution is slightly incremental. The main part of this work is the design of path consistency regularization.__
>
> __A__: First, we appreciate Reviewer 3wCP's recognition that
> 1) "the motivation is good",
> 2) "the method is easy to follow and the definition of path consistency loss is simple";
> 3) "empirical results on KITTI and NuScenes demonstrate the robustness of the proposed method."
>
> We also believe the merit of our proposed method is on its simplicity and effectiveness as we show the proposed method generalizes well in our empirical studies. We will release our code soon to facilitate reproducibility and adoption.
>
> Though the regularization is simple to use, it is the first time
> a regularization-based algorithm is applied to 3D detection and
> represent an orthogonal and complementary direction to the majority of
> existing works. As we pointed out, existing works often focus on 1) architecture changes (e.g., Deep-interaction, TransFusion, DeepFusion)
> or 2) new surrogate tasks (e.g., BEVFusion). In contrast, our regularization-based method provides a new model training perspective
> and shows that existing simple fusion methods are also capable of
> handling the semantic misalignment issue with proper regularization.
>
> __Q2: The forced alignment of the camera and LiDAR features is doubtful. Instead of aligning the features, learning an accurate transformation is more critical.__
>
> __A__: We would like to first clarify that we're not forcing
> 2D image features to be close to 3D point cloud features.
> Instead, our path consistency loss only depends on 2D features, forcing the __SAME__ 2D features to be transformed in a semantically consistent way in the 2D branch and 3D branch, respectively.
> As we described in Equation 3 in paragraph _Path-consistent regularization_ (page 5), both the _2D path_ and _3D path_ only rely on the same 2D input $x_t$.
>
> Meanwhile, the purpose of our path consistency regularization
> is consistent with Reviewer 3wCP's suggestion, that is, to learn a more accurate transformation from 2D to 3D.
> However, as we have shown empirically in Table 4 and Table 5,
> the network often cannot automatically learn a good transformation without proper regularization, so we introduce our method as a better transformation learning.
>
> Lastly, our path consistency loss only affects the parameters in the 2D backbone and lifting operators (see paragraph _Algorithm_ on page 6). And 2d backbone and lifting operator are largely responsible for learning an accurate transformation.
>
>
> __Q3: The description of the lift operator is unclear. Are the 2D features in different stages upsampled to the origin resolution before the lift transformation?__
>
> __A__: Yes, all lift operators will first upsample their corresponding 2D features to the same resolution (e.g., the original resolution).
>
> __Q4: Is that possible to adjust calibration parameters to directly lift 2D features to 3D features?__
>
> __A__: It is possible. But for this particular work,
> we followed PointPainting and FocalsConv to first upsample all 2D features to a fixed resolution and then directly use the calibration parameters for feature alignment.
> To ensure a fair comparison, we didn't explore beyond this recommended setting. Meanwhile, our focus is on demonstrating the value of our proposed path consistency loss, and hence, we didn't investigate different feature alignment strategies.
>
> __Q5: In practice, the point cloud and image cannot be aligned perfectly due to the synchronization error and motion distortion. For these cases, the shallow fusion is not reliable. Is deep fusion helpful in such situations?__
>
> __A__: We agree that a perfect alignment between 2D and 3D even at shallow layers is challenging.
> And we are not aiming for a perfect alignment between 2D and 3D in this work.
> Especially for our experiments on both KITTI and nuScenes, we applied GT-sampling to augment the point cloud as well as the image, which introduces distortion and synchronization errors.
> However, in alignment with our observation and prior work (e.g., Transfusion, PointAugmenting, FocalsConv), modern networks are often robust w.r.t. small misalignment and distortions.
> As long as 2D & 3D features are properly regularized, the network can benefit from additional semantic information from feature fusion and leads to improved downstream performance.

---

### Decision · Program_Chairs · 2023-01-20

**Decision:**

Reject

**Justification For Why Not Higher Score:**

Though this design choice makes sense, the empirical impact in performance is minor, as reviewers point out, and concerns a specific model architecture. The proposed design choice is motivated from a performance viewpoint in the introduction of the paper, and since its empirical validation is not convincing the reviewers and area chair find that it is not noteworthy enough to be published in ICLR.

**Justification For Why Not Lower Score:**

N/A

**Metareview: Summary, Strengths And Weaknesses:**

The paper addresses the problem of fusing 2D image frame features and 3D LiDAR features in a way that maximizes their alignment. The paper points out that in the input resolution there is an exact alignment between 3D scene locations and 2D pixel locations provided by camera calibration and the camera projection equations. However, during 2D and 3D featurization, this precise alignment is lost, and later fusion of 2D image features lifted in the 3D grid to be fused with LiDAR features is suboptimal due to the coarsening of 2D and 3D grids and their non precise alignments. The paper proposes multiple paths of fusing 2D image and LiDAR features in early or later layers, and imposes consistency losses between these paths, to encourage alignment of 2D and 3D features even at deeper layers. Though this design choice makes sense, the empirical impact in performance is minor, as reviewers point out, and concerns a specific model architecture. The proposed design choice is motivated from a performance viewpoint in the introduction of the paper, and since its empirical validation is not convincing the reviewers and area chair find that it is not noteworthy enough to be published in ICLR.

**Summary Of Ac-Reviewer Meeting:**

N/A